# ENHANCING HALLUCINATION DETECTION THROUGH NOISE INJECTION

**Litian Liu**[1]  **Reza Pourreza**[1]  **Sunny Panchal**[1]  **Apratim Bhattacharyya**[1]  **Yubing Jian**[1]
**Yao Qin**[2]  **Roland Memisevic**[1]
[1]Qualcomm AI Research* [2] UC Santa Barbara

## ABSTRACT

Large Language Models (LLMs) are prone to generating plausible yet incorrect responses, known as hallucinations. Effectively detecting hallucinations is therefore crucial for the safe deployment of LLMs. Recent research has linked hallucinations to model uncertainty, suggesting that hallucinations can be detected by measuring dispersion over answer distributions obtained from multiple samples drawn from a model. While drawing from the distribution over tokens defined by the model is a natural way to obtain samples, in this work, we argue that it is suboptimal for the purpose of detecting hallucinations. We show that detection can be improved significantly by taking into account model uncertainty in the Bayesian sense. To this end, we propose a very simple, training-free approach based on perturbing an appropriate subset of model parameters, or equivalently hidden unit activations, during sampling. We demonstrate that our approach significantly improves inference-time hallucination detection over standard sampling across diverse datasets, model architectures, and uncertainty metrics.

## 1 INTRODUCTION

Large Language Models (LLMs) have made significant advances in recent years (Achiam et al., 2023; Zhao et al., 2023). However, despite these advances, LLMs sometimes generate plausible yet incorrect responses – a phenomenon known as hallucination (Ji et al., 2023; Kuhn et al., 2023a)[1]. In light of this, effective hallucination detection during inference has gained significant attention and is essential for the safe deployment of current LLMs. One line of work detects hallucinations in a single sample by training a separate model (Azaria & Mitchell; Kossen et al., 2024; Liu et al.; Manakul et al., 2023; Su et al., 2024), allowing for evaluation on pre-defined question–answer benchmarks such as HaluEval (Li et al., 2023). This approach adds computational cost and it can suffer from train-test distribution shift Kossen et al. (2024). In contrast, we focus on an alternative line of work that detects hallucinations by assessing uncertainty across multiple samples *directly* drawn from the model (Chen et al., 2024; Kuhn et al., 2023a; Lin et al., 2024; 2022; Malinin & Gales, 2021;

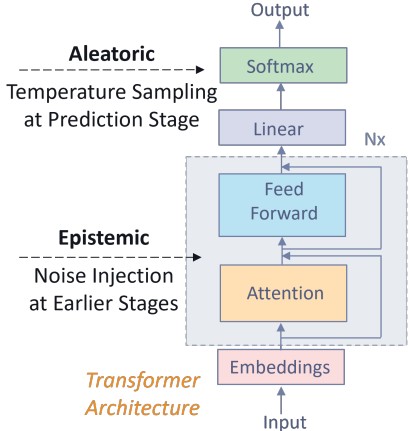

Figure 1: Inference-time hallucination detection typically relies solely on temperature-based sampling at the prediction layer, capturing mainly aleatoric uncertainty (Gao et al., 2024). We introduce noise injection to perturb intermediate representations. By combining noise injection with prediction layer sampling, our sampling approach captures both epistemic and aleatoric uncertainty.

---

*Qualcomm AI Research is an initiative of Qualcomm Technologies, Inc.

[0]Correspondence: `litiliu@qti.qualcomm.com`, `rmemisev@qti.qualcomm.com`

[1]Recent work (Kalai et al., 2025)) notes that hallucinations fundamentally stem from current training paradigms and may be inevitable without radical changes.

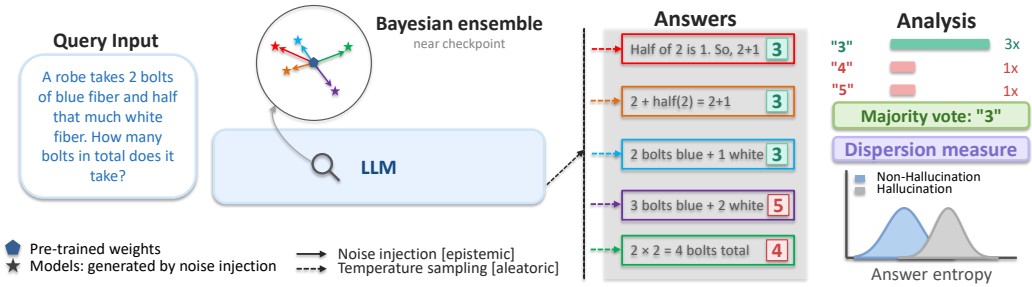

Figure 2: **Illustration of Noise-Enhanced Sampling for Hallucination Detection.** To sample an answer for a given input query, we first inject noise to sample a perturbed model from a Bayesian ensemble, then generate an answer from this model using temperature-based sampling, with both models and answers color-coded to indicate their pairing. Across samples, we measure the dispersion level (e.g., using answer entropy in Equation 4) to detect whether the majority-vote answer is a hallucination. This scheme captures epistemic uncertainty through noise injection and aleatoric uncertainty through temperature-based sampling, leading to improved detection compared to temperature-only sampling.

Manakul et al., 2023). For example, Lin et al. (2022; 2024) use semantic consistency and lexical similarity. Chen et al. (2024) quantifies uncertainty from the hidden activations of multiple samples. The core principle underlying this line of work is simple: the greater the observed uncertainty, the higher the likelihood of hallucination.

Since a language model defines the probability distribution over the next tokens, an obvious way to generate samples is to repeatedly draw from the conditional distribution over tokens given the context so far. This way of sampling stays faithful to the probability distribution defined by the model (up to any temperature-induced deviations from the training distribution), and it makes sense when the goal is to generate multiple answers, say, to a given prompt.

However, in the case of hallucination detection, the purpose of sampling is *not* to generate a diverse set of alternative answers to a given prompt. Instead, it is to estimate the coherence among sampled responses to a prompt, via a kind of "sensitivity analysis" that makes it possible to assess the likelihood of a given prompt to elicit a hallucination in a model. A distribution of responses that is coherent under perturbations is considered as evidence for the model knowing the correct response for a given prompt and accordingly, for the generated answers to be considered truthful.

More formally, sampling from the model using next-token prediction can be considered as a way to capture uncertainty in the data distribution, whereas to detect hallucinations, we are also interested in the model uncertainty, which is the result of training on a finite training set. The distinction between these two types of uncertainty has been studied formally by Osband (2016), who refers to the former as *aleatoric* (data uncertainty), and the latter as *epistemic* (model uncertainty).

This distinction is also reflected in a Bayesian perspective, where uncertainty over the model parameters reflects the epistemic uncertainty, and the model's output distribution reflects the aleatoric uncertainty. However, a full Bayesian treatment is challenging for LLMs, which contain billions of parameters and are trained on datasets that contain billions, and sometimes trillions, of tokens (Hou et al., 2024). It is also not feasible to apply approaches that cast dropout training as approximate Bayesian inference (Gal & Ghahramani, 2016b), since dropout is not included in many popular LLMs (see Appendix E). In this work, we devise a novel, simple yet effective *training-free* approach to approximate a surrogate distribution over models that are plausible given the training data, using pre-trained model weights as a starting point—as illustrated in Figure 1 and Figure 2.

To perform this approximation, we consider random perturbations of the parameters of a pre-trained model, which, as we show, is equivalent to perturbing hidden unit activations in some layers of the LLM for an appropriately chosen subset of parameters. Conveniently, the hidden activations also tend to capture the more abstract and high-level representations of a given phrase or "thought" (LeCun et al., 2015). This differentiates them from the output logits, which represent meaning at a

much lower, syntactic level, potentially making stability of hidden activations a better candidate to assess a model's faithfulness to the prompt in the context of detecting hallucinations.

Concretely, our surrogate distribution is uniformly distributed and centered at the pre-trained parameter weights of the hidden units and whose variance is defined by a single hyper-parameter. This retains the ability of models in the surrogate distribution to explain the training data well, while possessing sufficiently high coverage of plausible models to capture key aspects of the additional model uncertainty. This is illustrated in Figure 3, where we show the uncertainty associated with a prediction in the case of a hallucination, highlighting the effectiveness of jointly capturing epistemic and aleatoric uncertainty in a Bayesian framework.

In this work, we show how this insight leads to a very simple and efficient sampling approach to incorporate model uncertainty into inference-time hallucination detection. We demonstrate its effectiveness empirically across a wide range of datasets, model architectures, and uncertainty metrics.

## 2 BAYESIAN FRAMEWORK

Prior work (Chen et al., 2024; Kuhn et al., 2023a; Lin et al., 2024; 2022; Malinin & Gales, 2021; Manakul et al., 2023) connects hallucination detection to estimation of model uncertainty. Given an input context $\boldsymbol{x}$, the task of detecting hallucination can be formulated as a binary classification problem:

$$\mathcal{I}(\boldsymbol{x}) = \begin{cases} \text{Non-Hallucination} & \text{if } \mathcal{H}(\mathcal{Y}) < \tau \\ \text{Hallucination} & \text{if } \mathcal{H}(\mathcal{Y}) \geq \tau \end{cases}, \tag{1}$$

where $\mathcal{Y} = \{\boldsymbol{y}^1, \boldsymbol{y}^2, \ldots, \boldsymbol{y}^K\}$ denotes $K$ samples generated by the model given $\boldsymbol{x}$, $\mathcal{H}(\cdot)$ is an uncertainty metric, and $\tau$ is a detection threshold.

Prior work has primarily focused on designing uncertainty metrics, relying on standard sampling methods to remain faithful to the model's predictive distribution. Instead, we shift the focus to the sampling process and account for both aleatoric (data) and epistemic (model) uncertainty when drawing samples. To capture epistemic uncertainty, rather than relying on a single LLM, we consider the distribution of plausible models given the training data $\mathbf{D}$. Under this Bayesian formulation, the predictive probability of a sequence $\boldsymbol{y} = [y_0, y_1, \ldots, y_{t-1}]$, given input context $\boldsymbol{x}$ and the training data $\mathbf{D}$, can be expressed as (Malinin & Gales, 2021)

$$p(\boldsymbol{y}|\boldsymbol{x}, \mathbf{D}) = \int \prod_t p(\mathrm{y}_t|\mathrm{y}_{<t}, \boldsymbol{x}, \omega)p(\omega|\mathbf{D}) \, d\omega \,, \tag{2}$$

where $\omega$ denotes the model parameters and $p(\omega|\mathbf{D})$ is the posterior over parameters given the training data $\mathbf{D}$.

Since $p(\omega|\mathbf{D})$ is not directly accessible in practice, it is common to approximate it with a surrogate distribution $q(\omega)$. One approach to estimate $q(\omega)$ is to minimize the KL divergence between $q(\omega)$ and the true posterior $p(\omega|\mathbf{D})$ using a variational lower bound (MacKay, 2003; Gal & Ghahramani, 2016b). However, this approach remains challenging for large pre-trained LLMs, whose training data are often proprietary and whose retraining is computationally expensive. To address this, we propose a training-free surrogate distribution $q(\omega)$, which concentrates the parameter distribution around the pre-trained checkpoint while allowing controlled perturbations for a subset of parameters $\mathcal{S}$. Concretely, we define $q(\omega)$ as

$$q(\omega) = \prod_{i \notin \mathcal{S}} \delta(w_i - \bar{w}_i) \cdot \prod_{i \in \mathcal{S}} q_i(w_i \mid \bar{w}_i, \alpha) \,, \tag{3}$$

where $w_i$ represents the $i^{\text{th}}$ parameter of the model, $\bar{w}_i$ is the value of $w_i$ in the pre-trained checkpoint, and $q_i$ is a perturbation distribution centered around $\bar{w}_i$, with perturbation magnitude bounded by $\alpha$ to prevent unstable behaviors in the tails. For parameters in $\mathcal{S}$, we perturb according to $q_i$. For parameters not in $\mathcal{S}$, we fix them at their checkpoint values, effectively applying a Dirac delta distribution $\delta(\cdot)$ centered at the pre-trained value. Our choice of $q(\omega)$ is inspired by Kendall & Gal (2017), who show that distribution parameters concentrate as training data increases. Given the massive scale of LLMs, we expect this concentration to be even more pronounced, making a narrowly centered $q(\omega)$ a computationally efficient proxy for the true posterior.

Table 1: **Example of Answer Entropy Computation on the GSM8K dataset.** For each response, the answer string is marked in **bold**, with the remaining text representing the reasoning part. We estimate uncertainty by counting the occurrence of each answer string. In this example, with $K = 3$ samples, $\mathcal{H}_{ans}(\mathcal{Y}) = -0.67 \times \log 0.67 - 0.33 \times \log 0.33 = 0.275$.

| *Responses for* A robe takes 2 bolts of blue fiber and half that much white fiber. How many bolts in total does it take? | Answer | Answer Frequency |
|---|---|---|
| Half of 2 bolts of white fiber is 2/2 = 1 bolt. So, it takes 2 + 1 = 3 bolts in total. The answer is **3**. | 3 | 67% |
| 2 bolts of blue fiber and half that much white fiber is 2 + half of 2 = 2 + 1 = 3 bolts. The answer is **3**. | 3 | |
| 2 bolts of blue fiber and half that much white fiber is 2 x 2 = 4 bolts of blue fiber. The answer is **4**. | 4 | 33% |

In this work, we demonstrate the effectiveness of the Bayesian view with a simple and efficient approximation. Specifically, we restrict $\mathcal{S}$ to the bias terms in the MLP blocks, which have a similar effect to weight-based perturbations (see Appendix B). We implement the bias perturbation approximately via noise injection into the MLP activations. This design offers significant computational advantages: directly perturbing the model parameters would require a separate forward pass for each sampled model to generate every $y \in \mathcal{Y}$. In contrast, injecting noise into the activations allows independent noise per sample within a batch, enabling multiple models to be sampled and multiple outputs to be generated in parallel in a single forward pass. This preserves the Bayesian effect at a fraction of the computational cost.

Given that the activations are biased to be non-negative due to the SiLU nonlinearity in most models, we inject non-negative uniform noise into the activations. This corresponds to defining

$$q_i(w_i \mid \bar{w}_i, \alpha) = \mathcal{U}(w_i \mid \bar{w}_i, \bar{w}_i + \alpha)$$

which yields a naturally bounded perturbation to prevent unstable behaviors in the tails. Further details of our algorithm are presented through a case study in Section 3. As shown through extensive experiments in Section 4, this lightweight noise-injection approach substantially improves hallucination detection effectiveness. We explore alternative perturbation designs, including zero centered noise (Appendix H), a bounded Gaussian distribution (Appendix C), and perturbation of a different set of model parameters (Appendix B). All results corroborate the framework's effectiveness across these variations.

## 3 UNCERTAINTY AND HALLUCINATION DETECTION

In this section, we conduct a case study to investigate the effectiveness of the surrogate model distribution $q(\omega)$, as described above, in capturing epistemic uncertainty. We first hypothesize and validate that when sampling under the model distribution $q(\omega)$ (Equation (2)), the responses exhibit greater variability when the model hallucinates. We then observe that such epistemic uncertainty has a complementary effect when compared to aleatoric uncertainty for hallucination detection. Overall, combining epistemic and aleatoric uncertainty yields the best performance.

### 3.1 CASE STUDY SETUP

We perform an initial case study using the GSM8K dataset (Cobbe et al., 2021). Section 4 demonstrates that our algorithm also generalizes to knowledge-based question-and-answer tasks.

In this study, we use the GSM8K test set, containing 1319 questions, together with in-context learning examples from Wei et al. (2022). The dataset consists of mathematical question-response pairs $\{x, y\}$, where each response includes both the reasoning and the answer: $y = [r, a]$. As shown in Table 1, following in-context learning examples, an LLM can produce coherent yet incorrect answers—i.e., hallucinations—highlighting the need for effective hallucination detection in such reasoning tasks.

For effective hallucination detection for GSM8K through uncertainty estimation, we design an uncertainty metric as described in Equation (1). As illustrated in Table 1, reasoning chains can be

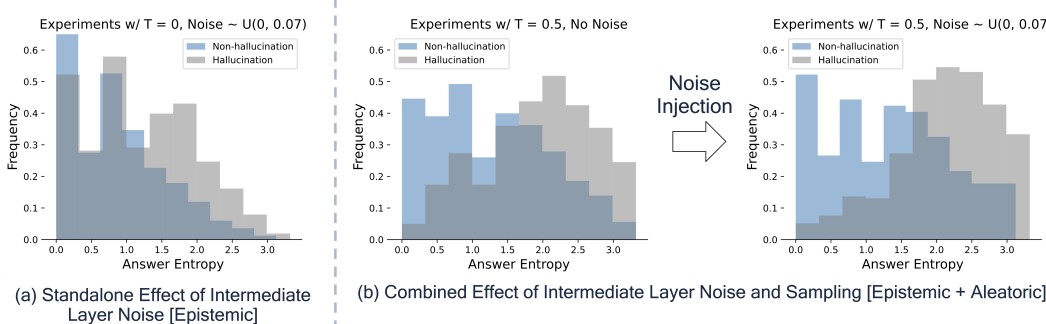

Figure 3: **Effect of Intermediate Layer Noise on Hallucination Detection.** *(a) Standalone Effect.* Noise injection induces epistemic uncertainty, where the LLM shows greater uncertainty for hallucinations (grey) than non-hallucinations (blue), as reflected by larger answer entropy (Equation 4). *(b) Combined Effect.* Combining noise injection with prediction layer sampling *(b Right)* improves hallucination/non-hallucination separation compared to using prediction layer sampling alone *(b Left)*, enhancing detection effectiveness. This highlights the importance of combining epistemic uncertainty with aleatoric uncertainty in sampling for hallucination detection. Evaluation on GSM8K with `Llama-2-7B-chat` model across 10 samples.

extensive, although the final answer holds greater importance. Consequently, assigning equal weight to all tokens during uncertainty estimation may be suboptimal. Since the final answer in GSM8K is numerical, metrics such as lexical similarity (Lin et al., 2022) or semantic entropy (Kuhn et al., 2023b) are less applicable. Instead, we estimate uncertainty by counting the number of occurrences of each final answer and introduce the metric of *Answer Entropy*:

$$\mathcal{H}_{ans}(\mathcal{Y}) = -\sum_j p(\mathrm{a}_j) \log p(\mathrm{a}_j) \tag{4}$$

where $p(\mathrm{a}_j)$ is the empirical probability of each unique answer $\mathrm{a}_j$ among the $K$ extracted final answers $\{\mathrm{a}^1, \mathrm{a}^2, \ldots, \mathrm{a}^K\}$ from responses $\mathcal{Y} = \{\hat{\mathrm{y}}^1, \hat{\mathrm{y}}^2, \ldots, \hat{\mathrm{y}}^K\}$. An example of the answer entropy computation is provided in Table 1.

In practice, for the GSM8K dataset in this section, and more generally for datasets with formatted answers (e.g., numeric responses or multiple-choice options), we count answer occurrences using exact string matching. For datasets with free-form answers, we instead group semantically equivalent responses using BERT embeddings, following the standard protocol of Li et al. (2020).

In the following, we focus on the `Llama-2-7B-chat` model (Touvron et al., 2023). Experiments with additional datasets, uncertainty metrics, and models are discussed in Section 4.

## 3.2 HALLUCINATION DETECTION UNDER EPISTEMIC UNCERTAINTY

We capture epistemic uncertainty through noise injection and study its effect on model hallucinations. Specifically, we inject uniform noise $\mathcal{U}(0, 0.07)$ to perturb the MLP activations of layers $20 - 32$ of the transformer. This approximately modifies the MLP bias and thus effectively samples a model $\hat{\omega}$ from our surrogate distribution $q(\omega)$. To isolate this effect, we set the prediction layer sampling temperature to zero and decode greedily to eliminate aleatoric uncertainty from sampling.

We generate $K = 10$ samples for each question and compute answer entropy following Equation (4). We classify model hallucination on a question level; model responses to a question are considered as hallucinating if the majority-vote answer from the $K = 10$ samples are incorrect, and as non-hallucinating otherwise. In Figure 3 (left), we compare answer entropy between hallucinating and non-hallucinating cases by overlaying the histograms of the two groups. We observe that with the model stochastically sampled from $q(\omega)$, responses exhibit greater variability when hallucinating (grey), as evidenced by higher entropy values. This shows the effectiveness of using noise injection for capturing epistemic uncertainty and thus detecting hallucinations.

We also remark that our experiments show a strong correlation between model reliability and the robustness of model output to perturbations. This aligns with findings in out-of-distribution (OOD)

Table 2: **Case Study: Effectiveness of Noise Injection for Enhancing Hallucination Detection.** With the same aleatoric uncertainty fixed by sampling temperature, noise injection (first row) introduces epistemic uncertainty, improving detection effectiveness over no noise (second row), as shown by a higher AUROC. Such improvement is achieved without degrading model generation accuracy (ACC). Evaluation on GSM8K dataset with `Llama-2-7B-chat` model across 10 samples.

|  | AUROC | ACC |
|---|---|---|
| Answer Entropy w/ $T = 0.5$, no noise | 71.56 | 23.64 |
| Answer Entropy w/ $T = 0.5$, noise $\sim \mathcal{U}(0, 0.07)$ | 76.14 | 24.09 |

detection research (Liu & Qin, 2025), where OOD test samples—i.e., samples with classes not seen during training—are flagged for eliciting unreliable outputs. In particular, Liu & Qin (2024) show that OOD samples often lie near decision boundaries and exhibit decreased robustness to perturbations compared to in-distribution (reliable) samples. Our results suggest that hallucinations in LLMs behave similarly, manifesting as measurable instability in model output under perturbation.

### 3.3 COMPLEMENTARY EFFECT OF ALEATORIC AND EPISTEMIC UNCERTAINTY

We now examine the interplay between aleatoric and epistemic uncertainty and their impact on model performance.

**Epistemic Uncertainty:** We inject noise sampled from $\mathcal{U}(0, 0.07)$ and set sampling temperature to zero as in Section 3.2.

**Aleatoric Uncertainty:** We set temperature as $T = 0.5$ and inject no noise. This inference scheme leverages the aleatoric uncertainty as captured by the original model.

For each setup, we assess answer entropy across $K = 10$ samples for each question following Equation (4). In the scatter plot in Figure 4, we display each question of the GSM8K test set as a point, with the x-value representing answer entropy under aleatoric uncertainty, and the y-value representing the same under epistemic uncertainty. The plot shows that model performance under the two types of uncertainty is weakly correlated, with a Pearson correlation of 0.58. This suggests that there is a positive but complementary relationship. We further validate the complementarity in Section 4.4.

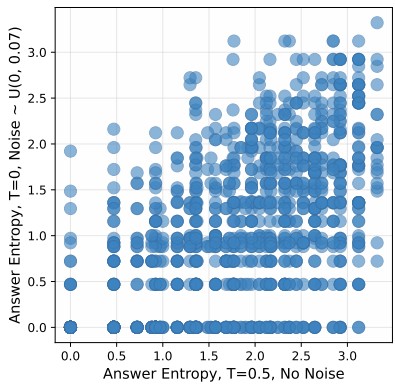

Figure 4: **Complementary Effect of Epistemic and Aleatoric Uncertainty**. The x-axis presents answer entropy (Equation 4) with prediction layer sampling only, mainly capturing aleatoric uncertainty. The y-axis presents answer entropy under intermediate layer noise injection only, mainly capturing epistemic uncertainty. A Pearson correlation of 0.58 indicates a complementary relationship between the two types of uncertainty.

### 3.4 NOISE-ENHANCED SAMPLING FOR HALLUCINATION DETECTION

To capture both epistemic and aleatoric uncertainty, as suggested by Section 3.3, we incorporate noise injection alongside prediction layer sampling and propose our noise-enhanced sampling for hallucination detection. The algorithm is illustrated in Figure 2 and described in Algorithm 1.

First, to capture epistemic uncertainty, we inject noise into MLP activations, effectively sampling models from our model distribution $\hat{\omega} \sim q(\omega)$ (see Equation (3)) as in Section 3.2. As LLMs include skip connections, adding independent noise across layers may cancel out; to prevent this, we instead use the same noise sample across all selected layers (see Appendix I for further discussion). Second, to capture aleatoric uncertainty, we sample from the temperature-adjusted categorical distribution $p(\mathbf{y}_t|\mathbf{y}_{<t}, \boldsymbol{x}, \hat{\omega})$. To detect hallucinations, we compute the answer entropy over $K$ samples and apply a threshold.

**Empirical Validation.** In Table 2, we validate the effectiveness of our scheme under the case study setup. We perturb the MLP activation of layers 20 to 32 with additive uniform noise of magnitude

---

**Algorithm 1** Noise Enhanced Sampling for Hallucination Detection

---

**input** input context $\mathbf{x}$, sample size $K$, uncertainty metric $\mathcal{H}(\cdot)$, model dimension $d$, temperature $T$, surrogate model distribution $q(\omega)$ (built from noise magnitude $\alpha$, perturbed layers $L_1$-$L_2$).
**output** Hallucination detection score: $s(\mathbf{x})$
 1: **for** $k = 1$ to $K$ **do**
 2:     `// Sample model from` $\hat{\omega} \sim q(\omega)$ `//`
 3:     Sample noise: $\epsilon \sim \mathcal{U}(0, \alpha)^d$
 4:     **for** each token $\hat{\mathbf{y}}_t^k \in \hat{\mathbf{y}}^k$ **do**
 5:       **for** each layer $l$ **do**
 6:         Compute $\mathbf{h}^l$ using the potentially perturbed prior layer representations.
 7:         **if** $l \in [L_1, L_2]$ **then**
 8:           Perturb the MLP activations: $\hat{\mathbf{h}}^l = \mathbf{h}^l + \epsilon$.
 9:         **end if**
10:       **end for**
11:       `// Sample tokens from model` $\hat{\omega}$ `//`
12:       Sample token $\hat{\mathbf{y}}_t^k \sim p(\mathbf{y}_t \mid \mathbf{y}_{<t}, \mathbf{x}, \hat{\omega})$ with temperature T.
13:     **end for**
14: **end for**
**return** Hallucination detection score $s(\mathbf{x}) = \mathcal{H}(\mathcal{Y})$, where $\mathcal{Y} = \{\mathbf{y}^1, \mathbf{y}^2, \ldots, \mathbf{y}^K\}$

---

$\alpha = 0.07$, sampled from $\mathcal{U}(0, 0.07)$, and evaluate over $K = 10$ samples. In practice, the noise magnitude can be selected based on the validation set, and we present an ablation study on different noise magnitudes in Section 4.4. Following Chen et al. (2024); Kuhn et al. (2023a); Lin et al. (2024; 2022); Malinin & Gales (2021), we assess the effectiveness of hallucination detection using the threshold-free metric, the area under the receiver operating characteristic curve (AUROC), where a higher value indicates better detection performance. As shown in Table 2, our scheme effectively detects hallucination instances with AUROC value $> 50$.

We further compare our approach with prior schemes that solely rely on prediction layer sampling without noise injection and thus do not capture epistemic uncertainty. The setup without noise injection follows Section 3.3. As shown in Table 2, our approach significantly improves detection effectiveness and achieves a higher AUROC value. The improvement is also visualized in Figure 3 *(b)*, where noise injection increases the separation and reduces the overlap in the histograms from left to right, significantly reducing high uncertainty hallucinations Simhi et al. (2025).

Finally, we evaluate model accuracy on the GSM8K dataset by applying majority voting on the generated samples; as before, we compare the aleatoric and epistemic settings. As shown in Table 2, taking into account epistemic uncertainty improves hallucination detection performance *without* degrading model generation accuracy, as indicated by the ACC column. We further analyze the dual improvements in generation accuracy and hallucination detection using a Pareto analysis across varying noise magnitudes in Appendix G.

Overall, our case study strongly supports our hypothesis regarding the relative importance of taking into account epistemic uncertainty in sampling.

## 4 EXPERIMENTS

In this section, we move beyond the case study to validate our algorithm across diverse datasets and architectures. We further conduct ablations to understand the effects of varying the number of samples, noise injection layers, noise magnitude, sampling temperature, and uncertainty metrics. As in Section 3.1, hallucination detection is evaluated using AUROC (see calibrated abstention analysis in Appendix F). Ablations are on `Llama-2-7B-chat` unless otherwise specified.

### 4.1 MAIN RESULT

In Table 3, we validate the effectiveness of noise injection for enhancing hallucination detection.

**Datasets:** Beyond the mathematical reasoning task GSM8K, we perform evaluations on CSQA (Talmor et al., 2019), which tests commonsense knowledge in a multiple-choice format, and TriviaQA (Joshi et al., 2017), which measures factual QA in a free-form setting. We use the CSQA validation set (1,221 questions) and the `rc.nocontext` subset of TriviaQA (18,669 questions).

Table 3: **Noise-Enhanced Sampling improves Hallucination Detection across Models and Datasets**. The gain shows the benefits of epistemic uncertainty alongside aleatoric uncertainty. Hallucination detection is evaluated with answer entropy, which applies across answer formats, using $K = 10$ samples. Detection AUROC is reported with mean and 95% confidence intervals. Higher mean values indicate better performance.

|  | GSM8K | CSQA | TriviaQA |
|---|---|---|---|
| Gemma-2B-it | 51.36 +/- 0.79 | 58.97 +/- 0.47 | 68.65 +/- 0.13 |
| Gemma-2B-it w/ Noise | 57.11 +/- 0.67 | 61.71 +/- 0.37 | 69.38 +/- 0.11 |
| Llama-3.2-3B-Instruct | 76.53 +/- 0.47 | 70.72 +/-0.49 | 77.40 +/- 0.07 |
| Llama-3.2-3B-Instruct w/ Noise | 82.70 +/- 0.34 | 72.83 +/-0.46 | 78.49 +/- 0.10 |
| Phi-3-mini-4k-instruct (3.8B) | 65.86 +/- 0.58 | 75.05 +/- 0.41 | 82.00 +/- 0.09 |
| Phi-3-mini-4k-instruct w/ Noise | 72.51 +/- 0.53 | 76.60 +/- 0.53 | 82.02 +/- 0.06 |
| Mistral-7B-Instruct-v0.3 | 75.85 +/- 0.36 | 76.52 +/- 0.36 | 75.86 +/- 0.11 |
| Mistral-7B-Instruct-v0.3 w/ Noise | 78.50 +/- 0.35 | 79.55 +/- 0.41 | 77.76 +/- 0.08 |
| Llama-2-7B-chat | 71.56 +/- 0.51 | 70.59 +/- 0.36 | 74.03 +/- 0.09 |
| Llama-2-7B-chat w/ Noise | 76.14 +/- 0.52 | 71.56 +/- 0.36 | 75.05 +/- 0.08 |
| Llama-2-13B-chat | 77.20 +/- 0.33 | 67.55 +/- 1.02 | 73.39 +/- 0.09 |
| Llama-2-13B-chat w/ Noise | 79.25 +/- 0.32 | 69.10 +/- 0.94 | 75.10 +/- 0.07 |

Table 4: **Noise-Enhanced Sampling Improves Hallucination Detection Across Diverse Uncertainty Metrics.** AUROC reported; higher is better. Evaluation on TriviaQA, whose free-form answers allow evaluation across all uncertainty metrics, using 10 samples.

|  | noise $= 0$ | noise $\sim \mathcal{U}(0, 0.09)$ |
|---|---|---|
| Predictive Entropy (Malinin & Gales, 2021) | 79.28 | **79.92** |
| Lexical Similarity (Lin et al., 2022) | 77.40 | **78.90** |
| Semantic Entropy (Kuhn et al., 2023b) | 75.70 | **77.21** |
| EigenScore (Chen et al., 2024) | 77.67 | **78.19** |
| selfCheckGPT-NLI (Manakul et al., 2023) | 75.80 | **77.53** |

Following standard protocols for hallucination detection Kuhn et al. (2023a); Chen et al. (2024), we generate plausible responses via in-context learning and treat incorrect answers as hallucinations (details in Section A.1). These datasets cover diverse topics, formats, and prompting styles. We adopt answer entropy as a unified uncertainty metric. For datasets with formatted answers (i.e., GSM8K and CSQA), answer occurrences for entropy computation (Equation 4) are obtained via exact string matching. For the free-form dataset TriviaQA, semantically equivalent responses are clustered in the BERT embedding space prior to frequency estimation, as discussed in Section 3.1).

**Model:** We evaluate a diverse range of LLMs across various sizes, including `Gemma-2B-it` Team et al. (2024), `Phi-3-mini-4k-instruct` (2.8B) Abdin et al. (2024), `Llama-3.2-3B-Instruct` Grattafiori et al. (2024), `Mistral-7B-Instruct-v0.3` (Jiang et al., 2023a), `Llama-2-7B-chat`, and `Llama-2-13B-chat`.

**Setup:** Following Section 3.1, we inject random uniform noise $\mathcal{U}(0, \alpha)$ into the MLP activation of upper layers. Since performance is not sensitive to specific layers (Section 4.5), we inject noise into roughly the top third of the layers as the default configuration (see Appendix A.2 for exact layer ranges). For the noise magnitude $\alpha$, we choose from $[0.01, 0.03, 0.05, 0.07, 0.09, 0.11]$ based on validation, as detailed in Appendix A.3. We fix the temperature at $T = 0.5$ and use answer entropy over $K = 10$ samples for hallucination detection. For each setup, we bootstrap 25 times from a total of 40 samples and report the mean AUROC with its 95% confidence interval.

**Performance:** Looking at Table 3, we observe that taking into account epistemic uncertainty consistently improves hallucination detection, as indicated by higher AUROC scores. The improvement is more pronounced on GSM8K and CSQA than on TriviaQA. This may be because GSM8K and CSQA involve chain-of-thought reasoning, where the model is called across multiple steps, effectively accumulating uncertainty. This is unlike TriviaQA, which relies directly on short

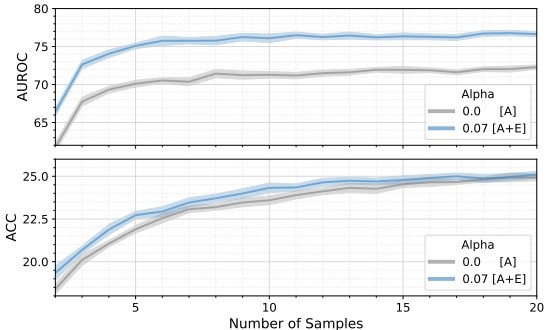

Figure 5: **Noise-Enhanced Sampling improves Hallucination Detection Across Different Numbers of Samples.** Evaluation with $T = 0.5$ for GSM8K using 1–20 samples. With $\alpha = 0$, only aleatoric uncertainty [A] is captured; with $\alpha = 0.07$, both aleatoric [A] and epistemic [E] uncertainty are captured. AUROC (upper) and ACC (lower); higher is better. Mean and 95% CIs shown.

|       | No Noise | $\alpha = 0.01$ | $\alpha = 0.03$ | $\alpha = 0.05$ | $\alpha = 0.07$ | $\alpha = 0.09$ |
|-------|----------|------|------|------|------|------|
| T=0.2 | 66.09 | 68.49 | 69.90 | 70.35 | 72.50 | 73.96 |
| T=0.5 | 71.56 | 75.71 | 72.73 | 74.06 | 76.14 | 75.24 |
| T=0.8 | 72.43 | 77.56 | 77.22 | 77.67 | 78.78 | 79.03 |
| T=1.0 | 72.34 | 77.12 | 77.68 | 78.34 | 78.14 | 78.22 |

Table 5: **Noise-Enhanced Sampling Improves Hallucination Detection Across Noise Magnitudes and Temperatures.** AUROC across temperature-noise combinations (darker is better). Temperature-only sampling plateaus (aleatoric limit), while noise introduces epistemic gains. Evaluation on GSM8K with 10 samples.

answers. Nonetheless, noise injection remains effective on TriviaQA. Notably, on TriviaQA with `Phi-3-mini-4k-instruct`, the baseline AUROC is already the highest across the board, suggesting performance saturation, which limits the impact of noise.

## 4.2 ABLATION ON UNCERTAINTY METRICS

Here, we show that noise-enhanced sampling is compatible with a variety of uncertainty metrics $\mathcal{H}(\cdot)$ from prior work, including Predictive Entropy (entropy normalized for sequence length) (Malinin & Gales, 2021), Lexical Similarity (based on Rouge-L scores) (Lin et al., 2022; 2024), Semantic Entropy (clustering similar texts before computing entropy) (Kuhn et al., 2023b), Eigen-Score (entropy in embedding space) (Chen et al., 2024), and selfCheckGPT-NLI (which measures discrepencies using the contradiction score from natural language inference) (Manakul et al., 2023).

We evaluate on TriviaQA, which has free-from answers and is compatible with all the uncertainty metrics. Table 4 reports AUROC for various metrics, at $T = 0.5$ and noise magnitudes $\{0, 0.09\}$. All metrics improve with noise injection, demonstrating the robustness of our approach.

## 4.3 ABLATION OF NUMBER OF SAMPLES

So far, we reported results using $K = 10$ samples. We now study how performance changes with different sample sizes. Figure 5 shows hallucination detection AUROC (left) and model accuracy (right) on GSM8K from $K = 1$ to $K = 20$, following the setup in Section 3.1. For each $K$, we report the mean and 95% interval across 25 bootstraps of $K$ samples from a total of 40 samples. Both AUROC and accuracy improve with more samples, and noise injection consistently enhances detection without degrading accuracy. In practice, $K$ can be tuned to the computational budget, but the benefit of noise injection holds across sample sizes.

## 4.4 ABLATION OF TEMPERATURE AND NOISE MAGNITUDE

In Table 5, we evaluate our algorithm under varying temperatures and noise magnitudes, following the setup in Section 3.1. While the optimal noise level depends on temperature, moderate injection consistently improves hallucination detection. The results also show that epistemic and aleatoric uncertainty are complementary: as temperature increases from $T = 0.8$ to 1.0 without noise, AUROC plateaus, but injecting noise at $T = 0.8$ boosts performance by capturing both sources of uncertainty.

## 4.5 ABLATION ON NOISE INJECTION LAYERS

We investigate the effect of noise injection across different layers, as examined on `LLaMA-2-7B-chat` (32 layers). Beyond the upper layers (20–32), we also inject noise into the middle (10–20), lower (0–10), and all layers (0–32). Table 6 reports hallucination detection AUROC for each setup. Noise magnitudes are set to $0.05, 0.01, 0.01$ for upper, middle, and lower

Table 6: **Noise injection across layers consistently enhances hallucination detection.** AUROC reported; higher is better. Evaluation on CSQA with 10 samples.

|  | AUROC |
| --- | --- |
| No noise | 67.55 |
| Lower Layer Noise | 70.03 |
| Middle Layer Noise | 69.68 |
| Upper Layer Noise | 69.10 |
| All Layer Noise | 70.68 |

Table 7: **Complementary Gain from Input Perturbation.** AUROC reported, higher is better. Evaluation on `Llama-3.2-3B-Instruct` with 10 samples. Best results achieved by combining both.

| input | model | GSM8K | CSQA | TriviaQA |
| --- | --- | --- | --- | --- |
| ✗ | ✗ | 76.53 | 70.72 | 77.40 |
| ✓ | ✗ | 76.10 | 72.07 | 78.33 |
| ✗ | ✓ | 82.70 | 72.83 | 78.49 |
| ✓ | ✓ | **82.84** | **72.98** | **79.20** |

layers, respectively, each selected from $0.01, 0.03, 0.05, 0.07, 0.09$ to achieve the best performance. As expected, lower layers require smaller magnitudes due to lower error tolerance for error propagation. For all-layer injection, we use noise magnitude $0.005$ to account for cumulative effects.

Table 6 shows that injecting noise into different layers of the model consistently improves performance compared to the baseline with no noise. This indicates robustness to layer choice and underscores the effectiveness of incorporating epistemic uncertainty into hallucination detection.

### 4.6 COMPLEMENTARITY GAIN FROM INPUT PERTURBATIONS

Jiang et al. (2023b); Gao et al. (2024) show that input perturbations can improve hallucination detection. These methods emphasize aleatoric uncertainty in the data, making them conceptually orthogonal and practically complementary to our method. In Table 7, we evaluate input perturbation on top of noise injection using `Llama-3.2-3B-Instruct`, where the former is implemented by shuffling in-context learning examples following Jiang et al. (2023b). The remaining experimental setup follows Section 4.1. We observe that combining both consistently yields the strongest results, confirming their complementarity.

## 5 RELATED WORK

**Bayesian Neural Networks.** Standard neural networks typically learn a single point estimate, neglecting epistemic and aleatoric uncertainty. Bayesian methods (MacKay, 1992; Neal, 2012) learn a posterior distribution over models to capture both uncertainty, but at a high computational cost. Gal & Ghahramani (2016b) addressed this using variational inference with a Bernoulli approximation of the weight distribution, subsequently extended to CNNs (Gal & Ghahramani, 2016a). For LLMs, Hou et al. (2024) argue that Bayesian methods are computationally impractical and instead quantify epistemic and aleatoric uncertainty using clarification questions. Here, we tackle this challenge with a novel, training-free Bayesian approach based on noise injection.

**Hallucination Detection.** As hallucinations cannot yet be fully eliminated, much research focuses on detecting them instead. A common strategy is to estimate model uncertainty across multiple samples (Lin et al., 2024; 2022; Manakul et al., 2023; Xiao & Wang, 2021; Kuhn et al., 2023a; Chen et al., 2024). Orthogonal and complementary to them, we introduce a sampling method that jointly captures epistemic and aleatoric uncertainty in a Bayesian framework. Another line of work avoids sampling by detecting hallucinations from a single inference Azaria & Mitchell; Kossen et al. (2024); Li et al. (2023); Liu et al.; Manakul et al. (2023); Marks & Tegmark; Su et al. (2024); Wang et al. (2025); Liu et al. (2026). While efficient at inference time, these methods typically require training auxiliary models on internal representations, adding computational overhead and falling outside the scope of sampling-based detection. Our work is also reminiscent of methods that perturb inputs instead of model activations (Hou et al., 2024; Jiang et al., 2023b; Gao et al., 2024), which however addresses aleatoric uncertainty in the data rather than epistemic uncertainty in the model, making them complementary to our approach (see Section 4.6).

## 6 CONCLUSION

This work tackles hallucination detection for the safe deployment of LLMs. While existing methods rely on aleatoric uncertainty through next-token sampling, we propose a very simple, *training-free* sampling approach that incorporates both aleatoric and epistemic uncertainty in a Bayesian manner. near its maximum likelihood parameters. Extensive experiments validate its effectiveness in improving hallucination detection in inference.

ACKNOWLEDGMENT

We sincerely thank Tongfei Guo (Northeastern University, ECE Department) for her contribution to the illustration in Figure 2.

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

# A    IMPLEMENTATION DETAILS

## A.1    DATASETS

We use in-context examples to demonstrate correct answer formatting and simplify answer extraction following free-form rationales, where applicable. For **GSM8K** and **CSQA**, we adopt the exemplars presented by Wei et al. (2022) as our in-context learning examples. The prompts guide the model to output the final answer after the anchor string "The answer is", as illustrated in Table 1, which we then extract accordingly for accuracy and answer entropy computation. For **TriviaQA**, we ensemble a 10-shot prompt from the first 10 training examples following Kuhn et al. (2023a).

On `Llama-2-7B-chat` and `Llama-2-13B-chat`, we concatenate in-context learning examples to form prompt using format Q:...A:...Q:...A:.... An example prompt for **TriviaQA** is:

```
Q: Which Oscar-nominated film had You Sexy Thing as its theme
song?  A: The Full Monty Q: Which Joan's career revived in
Whatever Happened to Baby Jane?  A: Crawford Q: Which much-loved
actor won the Best Actor Oscar for The Philadelphia Story?  A:
James Stewart (...)  Q: In which river is the Boulder Dam?  A:
```

If the model continues the Q:...A:... format after completing the answer, we trim generations using pattern matching with stopwords. For `Gemma-2B-it`, `Mistral-7B-Instruct-v0.3`, and `Phi-3-mini-instruct`, we apply the chat template available on the respective model tokenizers as available on Huggingface.

In evaluation, when the model fails to produce the answer with the correct format, we treat it as invalid.

## A.2    MODELS

All models evaluated in this work are off-the-shelf with no additional fine-tuning. We inject noise into roughly the top third of layers. Specifically, `Gemma-2B-it` has 18 layers in total, and we inject noise into layers 12-18. Similarly; for `Llama-3.2-3B-Instruct`, which has 28 layers, noise is injected into layers 20-28; for `Phi-3-mini-4k-instruct`, which has 30 layers, noise is injected into layers 20-30; for `Mistral-7B-Instruct-v0.3` and `Llama-2-7B-chat`, both with 32 layers, noise is injected into layers 20-32; and for `Llama-2-13B-chat` with 40 layers, noise is injected into layers 25-40.

Perturbations are implemented using the `IntervenableModel` interface of the open-source library `pyvene` (Wu et al., 2024), where we specify the injected noise, target layers, and intervention modules. The resulting `IntervenableModel` wraps the original model and seamlessly supports generation with noise injection. We run all of our experiments on 80GB NVIDIA A100s. And there is no noticeable latency overhead with or without noise injection, confirming that our method introduces no practical delay.

## A.3    NOISE MAGNITUDE SELECTION

We select the noise magnitude $\alpha$, based on the results over the validation datasets. For GSM8K, CSQA, and TriviaQA, respectively, for `Gemma-2B-it`, we set $\alpha$ as $0.05, 0.09,$ and $0.11$, for `Phi-3-mini-instruct`, as $0.05, 0.07,$ and $0.09$, for `Llama-3.2-3B-Instruct`, as $0.09, 0.09,$ and $0.07$, for `Mistral-7B-Instruct-v0.3`, as $0.03, 0.07,$ and $0.01$, for `Llama-2-7B-chat`, as $0.07, 0.03,$ and $0.09$, and for `Llama-2-13B-chat`, as $0.05, 0.05,$ and $0.09$.

Alternatively, noise magnitude can be selected per model by considering the signal to noise ratio (SNR). On `Llama-3.2-3B-Instruct`, we tune SNR on a combined validation set and inject noise from $\mathcal{U}(0, 0.3s)$, with $s$ being the activation magnitude. This improves detection AUROC for all datasets: GSM8K improves from 76.53 to 79.94, CSQA from 70.72 to 71.86, and TriviaQA from 77.4 to 78.35. Per-model selection offers a more efficient hyperparameter strategy than per-dataset tuning, remaining effective despite slight performance degradation.

## B  CONNECTION BETWEEN WEIGHTS AND BIAS PERTURBATION

We now show that injecting noise into the bias and the weight has a similar effect.

Consider an intermediate MLP layer with input $h^{in}$ and activation $h^{out}$. Assume the model is well-regularized, such that the input $h^{in}$ has a similar average magnitude $\sum_j h_j^{in}$ across samples. For computing the $i^{th}$ output element $h_i^{out}$, applying uniform noise into the corresponding weights $\theta_{i,j}$ is equivalent to injecting uniform noise from a rescaled magnitude into the layer's bias $\gamma_i$. Let $\epsilon$ be noise sampled from uniform distribution $\mathcal{U}(0, \beta)$. The output after injecting noise into the weights is computed as follows:

$$h_{out}^i = \sigma\Big( \sum_j \big( \theta_{i,j} + \epsilon \big) h_j^{in} + \gamma_i \Big) \tag{5}$$

$$= \sigma\Big( \sum_j \theta_{i,j} h_j^{in} + \sum_j \epsilon h_j^{in} + \gamma_i \Big) \tag{6}$$

$$= \sigma\Big( \sum_j \theta_{i,j} h_j^{in} + \big( \epsilon \sum_j h_j^{in} + \gamma_i \big) \Big), \tag{7}$$

where $\sigma(\cdot)$ is the activation function. By our well-regularization assumption, this is equivalent to perturbing the bias $\gamma^i$ with noise sampled from $\mathcal{U}(0, \beta \sum_j h_{in}^j)$.

## C  PERTURBING WITH BOUNDED GAUSSIAN NOISE: AN ALTERNATIVE BAYESIAN APPROACH

As an alternative instantiation of the Bayesian approach, we conducted an auxiliary experiment by injecting Gaussian noise with the same mean and variance as the uniform distribution. In Table 8, we experiment with `Llama-3.2-3B-Instruct` on CSQA, where hidden unit activations in layers 20–27 were perturbed with Gaussian noise bounded between $[0, \alpha]$ to prevent unstable behaviors in the tails. We sweep the noise magnitude $\alpha$ from $0.03, 0.05, 0.07, 0.09$. Our results imply that, under bounded perturbation, the hallucination detection improvements are not tightly coupled to a specific noise distribution, but rather to the perturbation dynamics governed by its statistical properties (i.e., mean and variance).

Table 8: **AUROC Performance Across Different Perturbation Distributions.** Perturbation with Gaussian noise (third row) performs comparably to perturbation with uniform noise (second row), as indicated by similar AUROC scores. Evaluation performed on the CSQA dataset with the `Llama-3.2-3B-Instruct` model across 10 samples.

| $\alpha$ | 0.03 | 0.05 | 0.07 | 0.09 |
|---|---|---|---|---|
| uniform | 71.14 | 71.23 | 72.32 | 72.83 |
| Gaussian | 71.19 | 71.24 | 72.3 | 72.64 |

## D  PERTURBING THE ATTENTION BLOCK: AN ALTERNATIVE BAYESIAN APPROACH

In this section, we explore an alternative instantiation of the Bayesian perspective by injecting noise into the attention block, as opposed to the MLP layer (see Figure 1). Specifically, we inject noise into the attention block activation, akin to modifying the unperturbed (zero) bias of the attention mechanism. In Table 9, we experiment with `Llama-2-7B-chat` on CSQA, perturbing the 20-32 layer activations with uniform noise. We sweep the noise magnitude $\alpha$ from $0.01, 0.03, 0.05, 0.07, 0.09$, and report the best performance at $\alpha = 0.01$. Our experiments show that this alternative perturbation achieves performance similar to MLP-activation perturbation, with both approaches enhancing hallucination detection. This further demonstrates the general effectiveness of Bayesian-inspired

noise injection in capturing both aleatoric and epistemic uncertainty, ultimately enhancing halluci-nation detection.

Table 9: **Analysis on Perturbation Position.** Noise injection at the attention activation (third row) performs comparably to injection at the MLP activation (second row), both improving detec-tion effectiveness compared to no noise (first row), as indicated by higher AUROC scores. This further demonstrates the general effectiveness of Bayesian-inspired noise injection in capturing both aleatoric and epistemic uncertainty. Evaluation was performed on the CSQA dataset with the `Llama-2-7B-chat` model across $K = 10$ samples.

|  | AUROC |
|---|---|
| No Noise | 67.55 |
| Noise Injection at MLP output | 69.95 |
| Noise Injection at Attention output | 69.10 |

## E   DROPOUT AND LARGE LANGUAGE MODELS

Many popular LLMs, such as the LLaMA family (Touvron et al., 2023), do not include dropout layers. Thus, the framework of Gal & Ghahramani (2016b), which casts dropout training as Bayesian approximation, does not extend naturally to LLMs. Nevertheless, we experiment with enabling dropout only at inference. Specifically, following the setup in Section 6, we randomly drop outputs of higher MLP layers with a 1% rate and observe a significant degradation in genera-tion accuracy (Table 10). In contrast, additive uniform perturbations preserve accuracy significantly better. Note that the model has not been trained under the additive noise condition. We hypothesize that additive noise is relatively benign, perturbing activations without entirely removing information. In contrast, dropout at inference can be more destructive, as dropping important units may disrupt critical information pathways and degrade performance sharply.

Table 10: **Dropout noise Degrades Generation Accuracy.** Model generation accuracy (ACC) is reported, with higher values indicating better performance. Evaluation was performed on CSQA across $K = 10$ samples.

|  | ACC |
|---|---|
| No Noise | 70.11 |
| Additive Noise | 70.83 |
| Dropout Noise | 67.50 |

## F   CALIBRATION VIA COVERAGE-ACCURACY ANALYSIS

To complement AUROC-based ranking evaluation, we assess calibrated abstention, which measures how well uncertainty scores guide selective prediction. Specifically, we compute coverage–accuracy curves for GSM8K on `Llama-3.2-3B-Instruct` in Figure 6, where coverage denotes the frac-tion of predictions retained after abstaining on the most uncertain cases, and accuracy measures correctness among the retained predictions. We sweep the coverage from 0.1 to 1 in steps of 0.1.

The results demonstrate that uncertainty scores derived from combined aleatoric and epistemic sources enable more effective selective prediction. In particular, the noise-injected setting achieves higher accuracy at low coverage and exhibits a steeper reduction in error as coverage decreases, indicating better calibration compared to aleatoric-only baselines.

## G   PARETO ANALYSIS UNDER NOISE INJECTION

To further investigate the dual improvements of model generation accuracy and hallucination detec-tion effectiveness noted in Sections 3.4 and Section 4.3, we evaluate a range of noise magnitudes

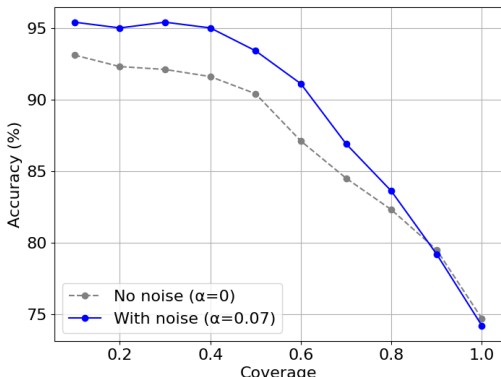 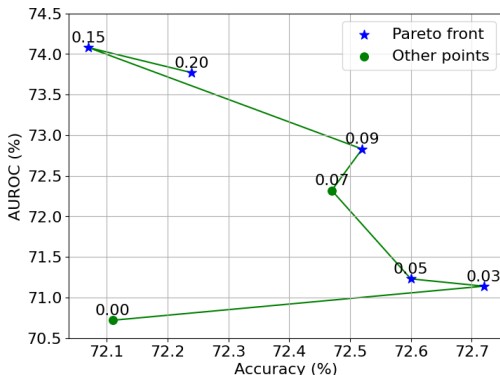

Figure 6: **Noise injection calibrated abstention.** Experiments conducted on GSM8K with `Llama-3.2-3B-Instruct`. Each point on the curve represents a coverage threshold, where the x-axis denotes the fraction of retained predictions after abstention and the y-axis reports the corresponding accuracy computed over those retained predictions. It shows that noise injection achieves higher accuracy at low coverage compared to the no-noise baseline, indicating improved uncertainty calibration.

Figure 7: **Pareto Analysis indicates Robustness to Noise Magnitude Selection.** Experiments on CSQA across 10 samples with `Llama-3.2-3B-Instruct`. Each point represents a noise magnitude $\alpha$, with x-axis showing generation accuracy (ACC) and y-axis showing hallucination detection (AUROC). Stars mark the Pareto front, highlighting noise magnitude that outperform the zero-noise baseline in both metrics, demonstrating robust gains across noise levels.

$\alpha = \{0, 0.03, 0.05, 0.07, 0.09, 0.15, 0.2\}$ for CSQA on `Llama-3.2-3B-Instruct`. Figure 7 visualizes the resulting Pareto front (starred points), representing noise magnitude that strictly outperform the zero-noise baseline $\alpha = 0$ in both metrics. The existence of this optimal region indicates that the benefits of noise injection are robust to the specific choice of magnitude within a moderate range.

## H  ZERO-MEAN NOISE INJECTION

In the main paper, we inject positive noise in MLP activations to preserve negative shift on activations. To further examine the impact of alternative noise variations, we evaluate three settings: no noise, positive-mean noise $\mathcal{U}(0, \alpha)$, and zero-mean noise $\mathcal{U}(-\frac{\alpha}{2}, \frac{\alpha}{2})$. Experiments are conducted on the CSQA dataset across $k = 10$ samples using `Llama-3.2-3B-Instruct`, with noise magnitudes $\alpha = 0.09$ as in Section 4.1.

Table 11: **Effect of Noise Mean on Performance.** Experiment on CSQA across 10 samples using `LLaMA-3.2-3B-Instruct`. AUROC and ACC reported, the higher the better.

|  | AUROC | ACC |
|---|---|---|
| No Noise | 70.72 | 72.11 |
| Positive Noise | 72.83 | 72.52 |
| Zero-Mean Noise | 72.27 | 72.22 |

As shown in Table 11, both noise-injection strategies yield consistent improvements over the no-noise baseline. Specifically, positive-mean noise achieves the highest AUROC of 72.83, while zero-mean noise closely follows with 72.27, compared to only 70.72 without noise injection. A similar trend is observed for generation accuracy ACC, where both positive and zero-mean noise outperform the baseline. Notably, we observe that zero-mean noise remains stable at even larger noise magnitudes, exhibiting no degradation in ACC.

Overall, these results suggest that the effectiveness of noise injection for hallucination detection does not critically rely on noise injection strategies. Instead, both positive and zero-mean noise improve model discriminability, indicating that the observed gains primarily stem from enhanced stochastic robustness.

Table 12: **Under skip connections in LLMs, shared layer-wise noise outperforms independent noise.** Experiment on GSM8K across 10 samples using `LLaMA-3.2-3B-Instruct`. AUROC reported, the higher the better.

|  | AUROC |
|---|---|
| No Noise | 76.53 |
| Shared Noise | 82.70 |
| Independent Noise | 80.20 |

## I NOISE DESIGN UNDER SKIP CONNECTIONS

Modern transformer architectures employ skip connections, which can cause independently injected noise across layers to partially cancel out as representations are aggregated. To mitigate this effect, we reuse the same noise vector across all selected layers rather than sampling noise independently per layer. Table 12 compares reused noise and independent per-layer noise on GSM8K using `LLaMA-3.2-3B-Instruct`, with noise sampled from $U(0, 0.09)$. Although independent noise improves over the noiseless baseline (80.20 vs. 76.53 AUROC), it underperforms relative to shared noise (82.70 AUROC). This result validates our design choice to inject shared noise across layers to mitigate cancellation effects.

