# OpenReview forum: "Enhancing Hallucination Detection through Noise Injection"
_ICLR.cc/2026/Conference — ICLR 2026 Poster_

### Official Review · Reviewer_fG6Y · 2025-10-16

**Soundness:** 2
**Presentation:** 3
**Contribution:** 2
**Rating:** 2
**Confidence:** 4

**Summary:**

The paper introduces an approach for uniform noise injection to LLMs which is motivated from a Bayesian point of view.
The authors use the approach for hallucination detection and show improvements over baselines on reasoning and QA tasks.

**Strengths:**

- The method improves over baseline hallucination detection methods on a range of benchmarks and clearly seems to separate hallucinated vs. non-hallucinated outputs better.

**Weaknesses:**

- I believe the paper is not precise enough about many of the theoretical concepts. First of all, the authors argue a lot for distinguishing between epistemic (connected to lack of knowledge) and aleatoric uncertainty (connected to inherent uncertainty in the problem).
However, they claim that "More formally, sampling from the model using next-token prediction can be considered as a way to capture uncertainty in the data distribution,"(L80-82), i.e. aleatoric uncertainty. This is arguably (as discussed in e.g. Depeweg 2018, Hüllermeier & Waegemann 2021) only the case if there is no epistemic uncertainty, because then the model has sufficiently learned the true distribution and the randomness in the outputs is indeed the randomness of the problem. But this is generally not the case for e.g. LLMs.
I am also surprised that the authors do not use any explicit decomposition of both uncertainties, which is discussed in these works and also in Malinin & Gales 2021, which is cited. Often Bayesian treatment is precisely motivated from this angle. The authors also motivated a Bayesian view but do not exploit it.

- Connected to this, the formulation of Eq. 2 is imprecise. It shows a token-level model averaging but it is not clear why this should be a better choice than sequence-level averaging (these are in general not the same). This is heavily discussed in Malinin & Gales 2021 but not picked up here. Furthermore, Alg. 1 also seems to rather approximate a sequence-level average as all full sequences are sampled using the same noise sample for each token within them if I understand the pseudo code correctly. As a consequence, it is also not clear to me how the baseline from their paper is calculated here: do you use an ensemble on a token-level or sequence-level or no ensemble at all?

- I believe claims that the claim "since LLMs are very expensive to train, Bayesian treatment of LLMs for hallucination detection has remained elusive" (L148-149) is not correct, as e.g. Shen et al. 2024 show Bayesian training of smaller LLMs using variational learning which could easily be used for hallucination detection.

- Regarding the experiments, the work does not compare to any other Bayesian methods like Laplace or variational learning, which can feasibly be done for LLMs also using LoRA. Therefore, it remains unclear how the uniform noise added here compares to related works and especially to learned noise which might turn out to perform better.

- For me it is unclear if the proposed answer entropy will work well outside of QA tasks, because it might be unlikely to sample the same (long and open-ended) answer twice. It would be good to add such an experiment if possible.

**Questions:**

- How exactly are the sequences y calculated in your algorithm? Is it correct that you are fixing the noise for a given input and then calculate the outputs by sampling from a model with the same noise applied for the entire sequence?

- Can you please provide details on how you apply the baseline from Malinin & Gales 2021?

- The citation Manakul et al. is broken.

- How would you calculate answer entropy for more open-ended tasks? It seems to me like there might very often be cases where all counts are 1 leading to maximal entropy just because it is much less likely to sample the same output many times when the output sequences are long and open-ended.

## References

Depeweg, Stefan, et al. "Decomposition of uncertainty in Bayesian deep learning for efficient and risk-sensitive learning." International conference on machine learning. PMLR, 2018.

Hüllermeier, Eyke, and Willem Waegeman. "Aleatoric and epistemic uncertainty in machine learning: An introduction to concepts and methods." Machine learning 110.3 (2021): 457-506.

Malinin, Andrey, and Mark Gales. "Uncertainty estimation in autoregressive structured prediction." ICLR, 2021.

Shen, Yuesong, et al. "Variational learning is effective for large deep networks." ICML, 2024.

---

> ### Author Response · Authors · 2025-11-21
>
> Thank you for your comments. We have corrected the citation for Manakul et al. Please see our responses to the remaining weaknesses and questions below.
>
> **Weakness 1: Uncertainty Analysis**  We agree that imperfect learning can bias the estimation of aleatoric uncertainty. Unfortunately, LLMs are not Bayesian models discussed in [2][3] and have only a single set of parameters, so standard sampling from an LLM can not explicitly measure epistemic uncertainty. Therefore, we follow the statement in prior work [1] that standard sampling mainly captures aleatoric uncertainty, as cited in Figure 1.
>
> Moreover, explicit decomposition of both uncertainties is not standard in hallucination detection [1]. For deciding whether to trust a response, an overall uncertainty estimate suffices, regardless of its explicit decomposition of aleatoric and epistemic sources. Meanwhile, we agree that such decomposition would provide valuable insights for future research.
>
> [1] Gao, Xiang, et al. "SPUQ: Perturbation-Based Uncertainty Quantification for Large Language Models." EACL. 2024.
>
> [2] Depeweg, Stefan, et al. "Decomposition of uncertainty in Bayesian deep learning for efficient and risk-sensitive learning." International conference on machine learning. PMLR, 2018.
>
> [3] Hüllermeier, Eyke, and Willem Waegeman. "Aleatoric and epistemic uncertainty in machine learning: An introduction to concepts and methods." Machine learning 110.3 (2021): 457-506.
>
> **Weakness 2 and Question 2: Token- vs. sequence-level averaging**  Our algorithm performs a sequence-level ensemble (Alg. 1), where each sampled sequence is generated under a fixed noise draw. Eq. 2 defines next token predictive probability, not sequences; so the token- vs. sequence-level distinction is not relevant there.  We added clarification in Line 147-152.
>
> We implement the uncertainty metric in [1] following prior work [2][3]. Specifically, we compute the Monte Carlo mean over length-normalized predictive probabilities. Implementation details are available in [3]. This metric is uniformly applied to standard sampling and our sampling; standard sampling has no model ensemble; our sampling reflects a sequence-level ensemble.
>
> [1] Malinin, Andrey, and Mark Gales. "Uncertainty estimation in autoregressive structured prediction." ICLR, 2021.
>
> [2] Kuhn, Lorenz, Yarin Gal, and Sebastian Farquhar. "Semantic Uncertainty: Linguistic Invariances for Uncertainty Estimation in Natural Language Generation." ICML 2023.
>
> [3] Chen C, Liu K, Chen Z, Gu Y, Wu Y, Tao M, Fu Z, Ye J. INSIDE: LLMs' Internal States Retain the Power of Hallucination Detection. ICLR 2024
>
>
> **Weakness 3 and 4 : Feasibility of training-based Bayesian LLMs and comparison**  The key contribution of our Bayesian approximation is that it is training-free, so direct comparison with training-based Bayesian methods would be unfair.
>
> Also, we disagree on the feasibility of training Baysian based modern LLMs. Training or finetuning to obtain an unbiased ensemble q(w∣D) requires the **full** pretraining dataset D of the LLM, accessible only to the original companies; none have released ensemble versions, making comparison infeasible. While LoRA based fine-tuning and variational methods [1] reduce training cost, they do not overcome the dataset constraints.
>
> [1] Shen, Yuesong, et al. "Variational learning is effective for large deep networks." ICML, 2024.
>
> **Weakness 5 and Question 4: Answer Entropy on Open-ended Answer**  Thank you for raising the concern.  We agree that answers that are identical as strings can be rare for open-ended answers. In fact, we did **not** compute Answer Entropy based on exact string matches for open-ended datasets such as TriviaQA. Instead, we group **semantically equivalent** responses using the standard protocol in [1]  based on BERT embedding similarity. This approach is widely used and naturally extends to longer, more open-ended responses. We have clarified this in the revised manuscript (see line 218-221).
>
> [1] Li, Bohan, et al. "On the Sentence Embeddings from Pre-trained Language Models."  EMNLP. 2020.

---

> > ### Comment · Reviewer_fG6Y · 2025-11-27
> >
> > Dear Authors,
> >
> > I would like to clarify a few points.
> >
> > > "Unfortunately, LLMs [...] have only a single set of parameters, so standard sampling from an LLM can not explicitly measure epistemic uncertainty."
> >
> > My review agrees with this but the point is that fixing the parameters does not take away the epistemic uncertainty unless it corresponds to the ''true'' parameters. This is also discussed in Wimmer at al. 2023 or Kotelevskii et al. 2025, for example, in Eq. 2. The part that is not modeled explicitly -but still there- is the excess risk (as a measure of epistemic uncertainty) due to the parameters not coinciding with those of the ''true'' data generating process.
> >
> > > "For deciding whether to trust a response, an overall uncertainty estimate suffices, regardless of its explicit decomposition of aleatoric and epistemic sources"
> >
> > This is in contradiction to the statement "to detect hallucinations, we are also interested in the model uncertainty, which is the result of training on a finite training set" (L83-85) which has motivated my point and clearly motivates such a decomposition, which from a Bayesian pov is possible by viewing $q(\omega)$ as an approximate posterior.
> >
> > > "Our algorithm performs a sequence-level ensemble (Alg. 1), where each sampled sequence is generated under a fixed noise draw. Eq. 2 defines next token predictive probability, not sequences; so the token- vs. sequence-level distinction is not relevant there."
> >
> > Thank you for clarifying this. However, what do you mean with "Eq. 2 defines next token predictive probability, not sequences; so the token- vs. sequence-level distinction is not relevant there.", could you make that more precise?
> > In my opinion the distinction is clearly relevant (as this modeling choice may lead to different estimates) and therefore it is also not clear to me why Eq. 2 motivates a token-level posterior but you use a sequence-level posterior. A token-level probability for a sequence-level posterior is found in e.g. Malinin & Gales Eq. 17.
> >
> > > "The key contribution of our Bayesian approximation is that it is training-free, so direct comparison with training-based Bayesian methods would be unfair."
> >
> > I think it would be easy to construct such a comparison, though, for example by finetuning with a variational optimizer (e.g. IVON from Shen et al. 2024) and comparing it to using finetuning with AdamW followed by your method or Laplace's method (cf. e.g. Yang et al. 2024 for a method used with LLMs).
> > In general I think the comparison to Laplace's method is highly relevant.
> >
> > > "In fact, we did not compute Answer Entropy based on exact string matches for open-ended datasets such as TriviaQA. Instead, we group semantically equivalent responses using the standard protocol in [1] based on BERT embedding similarity."
> >
> > Thank you for clarifying this point.
> >
> > ## References
> >
> > Kotolevskii et al., "From Risk to Uncertainty: Generating Predictive Uncertainty Measures via Bayesian Estimation", ICLR 2025
> >
> > Wimmer at al., "Quantifying Aleatoric and Epistemic Uncertainty in Machine Learning: Are Conditional Entropy and Mutual Information Appropriate Measures?", UAI 2023
> >
> > Yang et al., "Bayesian Low-rank Adaptation for Large Language Models", ICLR 2024

---

> > > ### Author Response · Authors · 2025-12-01
> > >
> > > Thank you for the additional comments. Please see our responses below.
> > >
> > > **Epistemic uncertainty**
> > > As noted in our original rebuttal, we agree that imperfect learning exists in LLM, but epistemic uncertainty is not modeled and thus cannot be explicitly quantified.
> > > Consequently,  standard sampling-based estimates capture aleatoric uncertainty with a bias introduced by imperfect learning. Following prior work [1], we therefore describe this estimate as mainly aleatoric.
> > >
> > > [1] Gao, Xiang, et al. "SPUQ: Perturbation-Based Uncertainty Quantification for Large Language Models." EACL. 2024.
> > >
> > > **composition of two uncertainties**
> > > We are not contradicting L83–85. For example, if the standard sampling-based method yields an uncertainty of 50 (reflecting biased aleatoric uncertainty) and our method yields an overall uncertainty of 100 (aleatoric + epistemic), then decisions such as thresholding are made on the overall uncertainty. The precise decomposition between aleatoric and epistemic (e.g., 70:30 vs. 60:40) is not important for this decision.
> > >
> > > **Eq 2 and token-vs-sequence level**
> > > We appreciate the reviewer’s point. Eq. 2 is not essential and may distract from the sequence-level posterior defined in Eq. 3 and Algorithm 1. We will remove Eq. 2 for clarity.
> > >
> > > **Bayesian Comparison**
> > > As noted in our original rebuttal, we disagree with the feasibility since finetuning an un-biased ensemble for SOTA LLM $p(w|D)$ requires **full training data** $D$, which are not publicly available. Fine-tuning methods do **not** remove the dataset constraints, as stated in our original rebuttal.

---

### Official Review · Reviewer_XrUM · 2025-10-30

**Soundness:** 3
**Presentation:** 3
**Contribution:** 3
**Rating:** 4
**Confidence:** 4

**Summary:**

The paper proposes a training-free, Bayesian-inspired sampling scheme for hallucination detection that perturbs a subset of model internals during generation to capture epistemic uncertainty, and combines it with standard temperature sampling that captures aleatoric uncertainty. Concretely, the method injects uniform noise into MLP activations (top layers by default), which is shown to be approximately equivalent to bias perturbation, and then samples answers K times; uncertainty is computed via Answer Entropy (and other metrics), yielding improved AUROC for hallucination detection across GSM8K, CSQA, and TriviaQA and multiple model families (Gemma, Phi-3, Llama-2/3, Mistral). Extensive ablations vary K, temperature, noise magnitude, and perturbed layers, and show complementarity with input perturbations.

**Strengths:**

1. Simple, training-free mechanism with negligible latency overhead; easy to bolt onto many LLMs and sampling-based detectors.
2. Principled motivation: separates aleatoric from epistemic uncertainty; implements the latter via internal perturbations approximating a posterior over models.
3. Consistent gains across datasets/models/metrics.
4. Thorough ablations.
5. Complementarity with input perturbations and with standard sampling - jointly using both sources yields the strongest results.

**Weaknesses:**

1. Noise distribution centering: The paper repeatedly injects U(0, α) (positive-only) noise and even defines​ q(omega) such that it is not centered at zero around the checkpoint parameter; this can introduce a mean shift rather than a purely variance-based epistemic perturbation. A symmetric zero-mean alternative is only partially explored and merits direct head-to-head comparison.
2. Metric dependence: A central result uses Answer Entropy (counting distinct final answers). This is natural for GSM8K/CSQA, but for long-form/free-form generation the mapping from perturbation to an “answer string” can be brittle; although the authors do show improvements for other metrics on TriviaQA, more analysis on long-form generation would strengthen claims.
3. Hyperparameter tuning burden: α is selected per model/dataset (and even per layer range). While inexpensive, this introduces an extra validation loop and raises questions about sensitivity/transfer across domains.

**Questions:**

1. Why not adopt zero-mean perturbations by default? Do your AUROC and accuracy trends hold when eliminating mean shifts?
2. How sensitive are results to reusing the same noise vector across layers vs. independent per-layer noise? Does structured, shared noise bias the uncertainty estimate?
3. Can you report calibration (e.g., coverage vs. error curves) to complement AUROC and show that epistemic perturbations improve calibrated abstention?
4. For long-form or multi-span answers (beyond a single numeric/string field), which aggregation metric worked best in your trials, and how robust is answer extraction?
5. What is the worst-case degradation in task accuracy across α, K, and perturbed layers? A Pareto curve (AUROC vs. accuracy) would clarify trade-offs.

---

> ### Author Response · Authors · 2025-11-21
>
> Thank you for your comments and suggestions. Please see our response below.
>
> **Weakness 1 and Question 1: Noise distribution centering**  We inject noise at the MLP activations (Line 181), where activations are mostly non-negative due to the non-linearity. Therefore, we use U(0,α) to avoid introducing negative values. As the induced noise is bounded by a very small α, the induced mean shift is very limited.
>
> In the meanwhile, we follow your suggestion and compare zero-centered  noise U(-α/2, α/2) and positive noise (U(0, α), as shown in Table 1 for CSQA on Llama3.2-3B with α=0.09. This validates that the two choices give similar performance in practice.
>
> **Table 1**.  Zero-centered noise and positive noise achieve similar performance.
> | Noise                    | AUROC | ACC   |
> | ------------------------ | ----- | ----- |
> | (U(0, α))           | 72.64 | 72.02 |
> | (U(-α/2, α/2)) | 72.27 | 72.22
>
> We also appreciate your comment on the definition of q(ω) and refined it in Section 2.
>
> **Weakness 2 and Question 4: Metric Dependence**  Thank you for raising the concern.  We agree that counting distinct strings for Answer Entropy is not an appropriate aggregation metric for multi-span datasets (e.g. TriviaQA).
> In fact, we did **not** count distinct strings for Answer Entropy on TriviaQA. Instead, we group **semantically equivalent** responses for answer entropy using the standard protocol in [1], based on BERT embedding similarity. This criterion is widely used and provides stable semantic grouping.  We have revised the manuscript to clarify this (Line 218-221).
>
> [1] Li, Bohan, et al. "On the Sentence Embeddings from Pre-trained Language Models."  EMNLP. 2020.
>
> **Weakness 3: Hyperparameter tuning burden**  Tuning cost can be further reduced by sharing a signal-to-noise ratio across datasets (Appendix A.3). While slightly below per-dataset optimal, this setting remains effective and demonstrates transfer across domains. Moreover, Section 4.5 shows that performance is largely insensitive to the choice of layer range, so searching α over multiple layer sets is not required in practice.
>
> **Question 2: Reusing noise** We reuse the same noise vector across layers to avoid noise cancellation induced by skip connections (Lines 170–171). As shown in Table 2, we compare shared noise across layers with independent per-layer noise. Although independent per-layer noise still improves over the noiseless baseline, its performance is worse than shared noise due to partial cancellation effects. Experiments are conducted on LLaMA-3.2-3B using GSM8K with noise sampled from U(0,0.09).
>
> **Table 2**. Independent noise improves over the noiseless baseline but performs worse than shared noise across layers.
>
> | Noise Type       | AUROC |
> | ---------------- | ----- |
> | Noiseless        | 76.53 |
> | Shared  Noise | 82.70 |
> | Independent Noise          | 80.20 |
>
> **Question 3: Calibration Report** Following your suggestion, we report calibration results on GSM8K using LLaMA-3.2-8B by measuring accuracy with coverage sweeping from 0.1 to 1 in steps of 0.1 in Table 3. Specifically, we compare the noise-injected setup (α=0.07) with the noiseless baseline (α=0). The results show that the noise-injected setting achieves higher accuracy at low coverage, indicating that epistemic perturbations improve calibrated abstention. Following your suggestion, we add the coverage-accuracy curve in Appendix F.
>
> **Table 3**: Noise injection improves calibrated abstention, achieving higher accuracy with low coverage.
>
> | Coverage                      | 0.1   | 0.2   | 0.3   | 0.4   | 0.5   | 0.6   | 0.7   | 0.8   | 0.9   | 1.0   |
> | ----------------------------- | ----- | ----- | ----- | ----- | ----- | ----- | ----- | ----- | ----- | ----- |
> | Accuracy (α=0)      | 93.1% | 92.3% | 92.1% | 91.6% | 90.4% | 87.1% | 84.5% | 82.3% | 79.5% | 74.7% |
> | Accuracy (α=0.07) | 95.4% | 95.0% | 95.4% | 95.0% | 93.4% | 91.1% | 86.9% | 83.6% | 79.2% | 74.2% |
>
> **Question 5: Accuracy-AUROC tradeoff** Task accuracy can reach near zero with large α, as shown in Table 4 (CSQA, LLaMA-3.2-3B).  Following your suggestion, we include AUROC vs. Accuracy curve in Appendix G to clarify trade-offs.  For visualization purposes, we omit α = 0.3 and α = 0 from the plot. The Pareto front includes α = 0.03, 0.05, 0.09, 0.15, 0.20, demonstrating improved trade-off compared to noiseless baseline.
>
> **Table 4**. Model accuracy significantly degrades with large α. The Pareto front (bold) demonstrates an improved AUROC–ACC trade-off compared to the noiseless baseline.
>
> | α            | 0     | **0.03**  | **0.05**  | 0.07  | **0.09**  | **0.15**  | **0.2**   | 0.25  | 0.3  |
> | ------------ | ----- | ----- | ----- | ----- | ----- | ----- | ----- | ----- | ---- |
> | Accuracy (%) | 72.11 | **72.72** | **72.60** | 72.47 | **72.52** | **72.07** | **72.24** | 53.20 | 0.04 |
> | AUROC (%)    | 70.72 | **71.14** | **71.23** | 72.32 | **72.83** | **74.08** | **73.77** | 19.88 | NA   |

---

### Official Review · Reviewer_Wqmu · 2025-10-31

**Soundness:** 3
**Presentation:** 3
**Contribution:** 3
**Rating:** 4
**Confidence:** 3

**Summary:**

There are many existing methods in literature for detecting hallucinations in an LLM that involve generating multiple samples from the LLM. This paper proposes adding noise to (some of) the parameters of the LLM during the sampling process. The authors argue that this noise injection is justified from a bayesian perspective and accounts for model uncertainty. The paper provides experimental results on the effect of adding noise to the intermediate layers of an LLM. The experiments section includes adequate comparisons to relevant baselines on established hallucination benchmarks, with noise injection showing favorable results.

**Strengths:**

- I like that the method is simple and justified from a theoretical perspective.
- The discussion on the complementary effect of aleatoric and epistemic uncertainty was insightful.
- The experiments section includes a variety of ablation studies which vary several relevant factors such as noise, temperature, layers in which noise is injected etc., Since, noise injection is easy to combine with other sample based approaches, I appreciate that the authors already included an ablation study that does this and it appears that injecting noise boosts the performance of those other methods.

**Weaknesses:**

Some choices and details were unclear to me and it would be nice to see more justification/discussion pertinent to these:
- Why did you pick the uniform distribution for the noise?
- How did you measure accuracy in Figure 4b? I am asking because there is no obvious way to judge the correctness of an LLM response compared to a ground truth. I believe it is important to be explicit about this detail (If you did include this in the manuscript already and I missed it, I apologize).

I am happy to raise my score if the above points and the questions I list below are sufficiently addressed.

Minor:
- Table 1 formatting: answer strings are not in bold as stated in the caption.

**Questions:**

- Is there a need to limit the layers for which noise is added? Maybe it does not matter that you add noise to all the model parameters as long as the magnitude (in some $L_p$ sense, is under some threshold). In light of the fact that the AUROC numbers in table 6 are all close to each other, I am more inclined to believe that this is the case.
- I understand you compare with Gaussian noise in Appendix C. But the AUROC performance is nearly identical with uniform noise. So does the noise distribution not matter? Why or why not?

---

> ### Author Response · Authors · 2025-11-21
>
> Thank you for your questions and comments. We have updated Table 1 with the answer string highlighted in bold. Please see our responses to the remaining weaknesses and questions below.
>
> **Weakness 1 and Question 2: Noise Shape** In principle, the choice of noise distribution matters: **unbounded distributions** can produce extreme perturbations that destabilize the model. Therefore, we use uniform noise because it is naturally bounded and avoids extreme perturbations. We updated the manuscript to clarify this in Line 172-173 & 181-183. In addition, we find that **bounded** noise distributions with mild shape differences yield similar results. For example, in Appendix C, we **bound** a Gaussian distribution to the same range as the corresponding uniform distribution, and they achieve similar AUROC performance.
>
> **Weakness 2: Accuracy Measurement in Figure 4b** In Figure 4b, we follow the standard protocol in [1] to measure GSM8K accuracy. Specifically, we (a) prompt the model with the standard chain-of-thought format (Line 664), so it outputs the final answer after the anchor string “The answer is” (see Table 1), and (b) extract this answer to compare with the ground-truth numerical value. We have added this explanation to Line 655-657.
>
> [1] Wei, J., Wang, X., Schuurmans, D., Bosma, M., Xia, F., Chi, E., ... & Zhou, D. Chain-of-thought prompting elicits reasoning in large language models. Neurips 2022.
>
> **Question 2: Layer Selection** We agree that performance is not sensitive to layer choice, as shown in Table 6. We restrict noise injection to upper layers mainly for efficiency: although lightweight, the cost scales linearly with the number of perturbed layers.

---

### Official Review · Reviewer_VoHR · 2025-11-02

**Soundness:** 3
**Presentation:** 3
**Contribution:** 3
**Rating:** 8
**Confidence:** 3

**Summary:**

The paper proposes introducing noise in LLMs during inference (by random perturbations of its parameters) as a way to capture epistemic uncertainty. Combined with temperature-based sampling, i.e., aleatoric uncertainty, the paper shows how the overall entropy is a better measure of uncertainty and thus does a better job in hallucination detection.

**Strengths:**

1. The paper focuses on a targeted research question, and does a good job of providing an in-depth discussion.
2. The improvement in hallucination detection is obvious. But for me, beyond these improvements, the paper provides a more accurate way to measure the overall uncertainty of an LLM, and thus, the technique here can be highly valuable.
3. The paper is well written, and I enjoyed reading it. The depth of the experiments is good, and the paper provides answers to many different ablation questions.

I should note that I'm not too familiar with the rapidly growing literature in hallucination detection, and separately, in LLM uncertainty measurement. So my assessment of the novelty of this work would be limited.

**Weaknesses:**

There are many other questions that could have been asked, but in my opinion, the paper did a good job focusing on one problem statement and providing appropriate depth.

My only complaint, and I hate to give such cliched feedback, but I really wanted to see experiments on a bigger variety of datasets. Different datasets seem to have different trends in the 3 datasets used in the paper, and an analysis over a larger set of datasets would have been very interesting. For instance, how well does this technique work for factual hallucinations vs faithfulness hallucinations? How well is it able to handle knowledge-intensive hallucination datasets vs commonsense reasoning datasets? Etc.

**Questions:**

1. Based on the trends in Figure 2, as well as the fact that improvements by adding epistemic uncertainty calculations are not extreme (although they are clearly substantial), it seems that epistemic uncertainty plays a smaller role in hallucination detection than aleatoric uncertainty. Can you comment on why you think that might be? Is the expectation that there would be less epistemic uncertainty? Or is the problem instead that the approximation is not good enough, and future work should focus on improving how we can measure epistemic uncertainty?
2. Input perturbations should be a form of epistemic uncertainty, not aleatoric uncertainty as mentioned in the paper. I would like to hear the authors' reasoning if they disagree. Assuming it is a form of epistemic uncertainty (which I believe it is), why do input perturbations have a complementary effect to an existing measure of epistemic uncertainty? This probably connects to my first question, but a broader discussion on what is lost when trying to 'approximate' epistemic uncertainty, since we cannot perform a proper Bayesian treatment of LLMs, and how it can be supplemented with other additions, would help the paper.

---

> ### Author Response · Authors · 2025-11-21
>
> Thank you for supporting our paper! Please see our responses below.
>
> **More Datasets**  We aimed to cover several distinct hallucination types: GSM8K captures faithful hallucination in reasoning, TriviaQA captures factual hallucination, and the contrast between CSQA and TriviaQA reflects commonsense vs. knowledge-intensive settings. We agree that many additional datasets could be explored, and extending our analysis to a broader range would be a valuable direction for future work.
>
> **Epistemic Uncertainty** Epistemic uncertainty is likely not fully explored under our scheme. Specifically, since we observe that the performance collapses when slightly larger noise is added, our scheme is constrained to using very small noise. This limits the amount of epistemic uncertainty we can probe. Developing better approximations is therefore a valuable direction for future work.
>
> **Input Perturbation**  We remark that prior works interpret uncertainty from input perturbation from both perspectives: some [1] as epistemic, others [2] as reflecting input ambiguity, i.e., aleatoric. We follow the latter view, which matches our observations: input perturbations have little effect on GSM8K, where questions are mathematical and unambiguous, but more effect on TriviaQA.
>
> [1] Gao, Xiang, et al. "SPUQ: Perturbation-Based Uncertainty Quantification for Large Language Models." EACL. 2024.
>
> [2] Hou B, Liu Y, Qian K, Andreas J, Chang S, Zhang Y. Decomposing uncertainty for large language models through input clarification ensembling. ICML 2024.

---

> > ### Comment · Reviewer_VoHR · 2025-11-27
> >
> > The authors' response is appreciated, and the rebuttal is acknowledged. I continue to be on the positive side for this paper.

---

> > > ### Author Response · Authors · 2025-12-01
> > >
> > > Thanks again for your time and thoughtful feedback! We really appreciate your support.

---

### Comment · Area_Chair_39u1 · 2025-11-27
**Rebuttal and Discussion Phase**

Dear Reviewers,

Thank you again for your time and effort in reviewing this paper. We are approaching the discussion deadline. I kindly ask you to review the rebuttal and continue the discussion so that we can reach a well-considered decision.

---

### Meta-Review · Area_Chair_1gs2 · 2026-01-08

**Summary:**

Instead of uncertainty over samples, they want to perturb model parameters during inference for hallucination detection.

Based on the reviews, rebuttal and follow up responses, I am able to determine that the authors were able to partially address concerns about experimental design and further expand on the analysis. However, they were not able to fully satisfy some of the reviewer concerns about the central assumptions of the work or fully compare to other related work, which I agree with the authors might be too expensive to be feasible.

However, I am not confident in this assessment, which makes this paper a borderline accept for me, where I’m okay with the SACs bumping down my decision.

**Reviewer Concerns:**

1. Wqmu: questions about details of the implementation, including noise distribution assumption and inject design. I believe the authors adequately answer these questions.
2. Xrum: hyperparameter choices, evaluation on long-form generation and not just exact string matching, noise distribution, calibration.
3. Wqmu, xrum, fg6y: Evaluation metrics for triviaqa.
4. Fg6y: token vs sequence level aggregation.

**Unaddressed**
1. Vohr, fg6y: Too few datasets, and trends are in all different directions, so unclear how generizable the results are.
2. Vohr, fg6y: issues with the central theoretical assumption in the work. The paper assumes that the central method in the paper, i.e. input perturbations represent aleatoric (inherent) uncertainty in models, and NOT knowledge-based or epistemic uncertainty. This seems counterintuitive, as perturbing the inputs seems directly related to perturbing the knowledge (epistemology). The authors’ response seems to not really address this critique.
3. Fg6y: the work does not compare to any other Bayesian methods like Laplace or variational learning, which involve training. The authors respond by saying that this comparison would be unfair

**Reviewer Scores:**

- Vohr: Not confident in their own rating, but continued to support the acceptance of the paper, after rebuttal.
- wqmu: I would guess that the author response satisfied this reviewer’s objections, and if they had responded, they would have increased their score.
- xrum: suggests many different analyses to probe the design in the paper, which to the best of my assessment, the authors provide empirical evidence for, along with expository responses.
- fg6y: Based on the follow up response, it’s unclear if this would results in an increase of their scores beyond a 4.

---

### Decision · Program_Chairs · 2026-01-26

Accept (Poster)